# Effects of Authentic Leadership on Intrapreneurial Behaviour: A Study in the Service Sector of Southern Spain

**DOI:** 10.3390/bs14080705

**Published:** 2024-08-12

**Authors:** Alejandro González-Cánovas, Alejandra Trillo, María Magdalena Jiménez-Barrionuevo, Francisco D. Bretones

**Affiliations:** Faculty of Labour Relations and Human Resources, University of Granada, 18071 Granada, Spain; alegonzalez@correo.ugr.es (A.G.-C.); atrillo@ugr.es (A.T.); marilena@ugr.es (M.M.J.-B.)

**Keywords:** authentic leadership, intrapreneurship, meaningful work, practised creativity, work autonomy

## Abstract

Employees currently face an increasingly demanding environment in terms of intrapreneurial behaviour due to the key role it plays in the survival of companies and the elimination of threats in the organisational environment. This paper investigates the antecedents of intrapreneurial behaviour in the service sector in southern Spain, analysing the relationships between it and authentic leadership, as well as the mediating role of practiced creativity, autonomy and meaningful work. This study uses a quantitative approach through the distribution of a questionnaire. A total of 333 employees completed the research questionnaire. The results show that although authentic leadership has a significant negative direct effect on intrapreneurial behaviour, it positively and significantly promotes it through practiced creativity and meaningful work. In contrast to the previous two mediations, the mediation of autonomy was not significant. This research provides empirical findings that can contribute to a better understanding of intrapreneurial behaviour by highlighting the importance of balanced, authentic leadership and offering guidance to organisations in designing a work climate that enhances organisational effectiveness.

## 1. Introduction

The role of employees in organisations has changed in ways that encourage them to be increasingly proactive and innovative, take on more responsibility and to decentralise decision making [1]. In short, today’s organisations are increasingly asking their employees to adopt a more intrapreneurial way of working to meet, or even initiate, these changing demands.

For decades, entrepreneurship, defined as the business endeavours of individuals operating outside the context of existing organisations, has been primarily associated with the creation of a company or venture [2]. However, entrepreneurship is fundamentally an attitude as well as a behaviour and perspective that shapes one’s understanding of their environment and consequent actions [3]. In this context, there is an increasing demand in organisational settings for employees who exhibit intrapreneurial behaviour [4] because of their contribution to innovation, performance and economic growth.

In this context, the term intrapreneurial behaviour (IB) refers to the process by which an employee identifies and exploits opportunities through innovation, proactivity and risk-taking to develop new products, services or processes, thereby improving competitiveness and organisational performance [5]. Unlike traditional entrepreneurship, IB does not mean that employees leave their organisations or risk their capital to pursue their ideas independently; rather, they use the organisation’s resources to innovate and drive change within the organisation [6]. In this way, IB occurs when employees, characterised by an anticipatory vision, develop new innovative ideas or practises that add value to the organisation and have a positive impact [7].

As a result, IB has become a critical skill for the success of today’s companies because it enables organisations to identify business opportunities, develop innovative solutions and implement them in the work context [8]. However, despite the growing attention in both the academic and management literature to the mechanisms for fostering IB and the variables that can promote this orientation in employees, such as leadership style and the promotion of meaningful work [9,10], there remains a gap in understanding how these variables specifically interact in the context of authentic leadership.

Therefore, our aim in this study is to analyse the variables of practised creativity (PC), meaningful work (MW), authentic leadership (AL) and intrapreneurial behaviour (IB) and to propose an integrated model of IB. Thus, this study makes several contributions.

First, our study extends the understanding of IB from an individual perspective, an area that is less well studied than the more business-oriented approach to entrepreneurial behaviour. Second, our work analyses which variables are associated with intrapreneurial behaviour, providing a more complete picture. Finally, this study has practical implications by enabling managers to implement human resource policies and practises that promote intrapreneurial behaviour among their employees.

## 2. Theoretical Background and Hypothesis Development

The study of intrapreneurial behaviour has gained relevance in recent years because of the growing number of employees who possess knowledge and skills and whose entrepreneurial spirit is not oriented towards the creation of new enterprises but who have an ideal profile to become intrapreneurs [11]. Intrapreneurship is a bottom-up innovation that refers to the innovative behaviour of employees within firms when making decisions and developing novel solutions to business problems and market changes [12]. This relevance is due to the identification of intrapreneurial outcomes as critical to organisational performance and growth measures. In addition, the behaviour of intrapreneurial employees generates initiatives that can multiply and influence teams’ performance [13,14]. Therefore, as some authors have pointed out [15], it is essential to implement organisational policies that promote IB among its employees because if the organisational culture is resistant to change and reluctant to experiment with new ideas, it could discourage intrapreneurship.

Several studies have found that the antecedents with the greatest impact on IB are both organisational and employee characteristics [16,17]. One of the organisational variables is the influence of the manager. There are studies in the literature that have analysed the influence of different types of leadership styles on the development of intrapreneurial behaviour among employees [18,19,20]. Currently, there are several leadership styles. Traditionally, the dominant leadership model has been that of transformational leadership developed by Bass [21]. However, more recently, other models have been developed, such as ideological, servant, spiritual, or authentic leadership [22]. Of these, authentic leadership (AL) is perhaps the most developed in the empirical literature, largely due to the development of validated measures and theory [23,24]. 

However, although there is some controversy between authentic leadership and other leadership styles, some authors believe that authentic leadership can serve as a “root” [25,26], focusing more on the personal characteristics of leaders rather than their processes or behaviours, as is the case with transformational leadership [27].

Thus, in the case of authentic leadership (AL), the model suggests that both leaders and followers have a moral obligation to look after the interests of the collective [28], which would explain the stronger relationship of AL with citizenship and collective behaviours, although these behaviours do not always produce tangible individual rewards, unlike the transformational leadership style, which is more focused on achieving results [29].

In this context, the theory of authentic leadership developed by Avolio et al. in 2004 [30] is particularly useful. This theory emerges in response to the need for leadership characterised by moral integrity and the ability to inspire trust within organisations, particularly in contexts characterised by change. According to this theory, authentic leadership, defined as a pattern of leadership behaviour that promotes positive capacities for greater self-awareness, an internalised moral perspective, balanced information processing and relational transparency, fosters positive organisational self-development, such as perceptions of justice or organisational commitment, among others [31,32]. 

Another theory that can enhance the understanding of how authentic leaders influence employee behaviour is the social information processing (SIP) theory [33]. According to this theory, employees interpret and respond to their work environment based on the social cues and information they receive from leaders and co-workers. Thus, by creating a social context based on transparent and ethical behaviours, authentic leaders provide clear and consistent social information that employees use to shape their perceptions and attitudes towards work, thereby influencing their intrapreneurial behaviour.

For example, several authors [34,35] have found that authentic leadership is also better at fostering creativity and innovation. Therefore, by focusing on a more internalised moral perspective beyond performance or results, followers perceive their work environment as more conducive to trying new things, which is a good precursor to intrapreneurial behaviour [20].

Previous research [36] has revealed that when leaders exhibit these behaviours, followers tend to be more creative, challenge standard work practises and are more likely to persuade their managers to implement more innovative ideas within the organisation. This is because employees receive good feedback that does not censor their intrapreneurial thoughts and are willing to take risks because they perceive that their leaders want to see them perform new tasks and projects effectively [37]. In other words, by fostering an environment of trust, respect and identification among followers, they facilitate the generation of new ideas and strengthen the perception of freedom to take risks and propose contradictory beliefs without fear of rejection, despite the challenges of continuous change in the business environment [38]. Authentic leaders, through their leadership behaviours, could therefore serve as a catalyst for generating IB among subordinates by creating a work environment conducive to innovation and creativity [20]. Therefore, with reference to previous studies, such as that by Cai et al. [39], our first working hypothesis is as follows: 

**H1:** 
*Authentic leadership (AL) is positively associated with employees’ intrapreneurial behaviour (IB).*


However, the relationship between AL and IB may be mediated by the personal resources that employees possess. As discussed above, contemporary organisations devote considerable resources to promoting the creativity and high-quality ideas of their employees [40]. Creativity is the ability to generate novel and appropriate ideas aimed at solving problems or improving organisational efficiency [41].

Several studies [42] have suggested that AL can play a crucial role in promoting creativity (PC). In particular, it has been argued that if an authentic leader creates an environment characterised by fair and transparent interactions, employees will be more willing to experiment with new ideas [43,44]. In this line of thinking, the authors of [38] highlight that employee creativity increases because of the perception of psychological safety and the increase in intrinsic motivation promoted by authentic leadership. Thus, although creativity and intrapreneurship are independent constructs, creativity is the most important characteristic of entrepreneurs. In this context, several authors [45,46] argue that PC is a crucial facilitator of successful innovation. This, in turn, serves as a precursor to IB, as it enables employees to identify opportunities, develop innovative solutions, and adapt to changes in the business environment.. Therefore, based on the literature reviewed, the following hypothesis is proposed:

**H2:** 
*Perception of practised creativity (PC) mediates the relationship between authentic leadership (AL) and intrapreneurial behaviour (IB).*


Similarly, to better understand the relationship between AL and organisational behaviour, some authors [47] have studied the interaction between this leadership style and employees’ propensity for autonomous work (WA). This interest is based on the observation that authentic leaders cultivate a work environment characterised by a higher degree of individual freedom, self-confidence and growth opportunities for their team members [48]. Within this framework, WA is postulated to derive from authentic leadership because it enables employees to make more effective use of their domain-specific skills and knowledge. Indeed, some authors [49] have highlighted the importance of a work environment that promotes autonomy, noting that a work environment that promotes autonomy attracts and retains creative talent and motivates employees in their work tasks. Since then, the relationship between WA and IB has been the subject of research [50,51]. These studies have shown a positive relationship between high WA levels and increased IB generation within an organisation. WA may also influence the association between AL and IB, as it catalyses innovative thinking and induces extra-professional behaviours that benefit the organisation eventually [52,53].

Therefore, based on previous studies, such as the one conducted by Alam et al. [54] that demonstrated the mediating role of autonomy in the relationship between other organisational factors and intrapreneurship, we propose the following hypothesis:

**H3:** 
*Work autonomy (WA) mediates the relationship between authentic leadership and intrapreneurial behaviour (IB).*


However, the relationship between authentic leadership and the development of intrapreneurial behaviour may also be mediated by the employee’s meaning of work (MW). Several studies have shown that the design and meaning of employees’ work tasks are influenced by the behaviour of their supervisors [55]. As authentic leaders create a sense of community belonging through the articulation of a shared vision and goals, the existential meaning attributed to work activity is enhanced [56]. Furthermore, authentic, altruistically motivated leadership promotes a greater sense of belonging by strengthening one’s identification with the organisation and providing greater social support [57]. In support of this perspective, Chaudhary [58] found that the meaning attributed to work (MW) is the result of empowerment fostered by authentic leaders through a greater sense of responsibility and value. 

Similarly, individuals who attach high levels of meaningfulness to their work tend to exert extra effort, leading to increased innovativeness [59]. Thus, the perception of work as intrinsically meaningful drives the use of skills and resources to realise innovative performance [60]. Based on recent research [61] showing that a greater sense of meaning at work mediates the relationship between leadership and higher levels of innovative effort (associated with intrapreneurial behavior), we propose the following hypothesis: **H4:** Meaningful work (MW) mediates the relationship between authentic leadership and intrapreneurial behaviour (IB).

In summary, considering IB and the relationship between several variables, we propose the following research model, as shown in Figure 1.

## 3. Research Methods

### 3.1. Sample and Data Collection 

To test each of the hypotheses, we conducted a survey of a sample of employees at three service companies in Andalusia, a region in southern Spain known for its dynamic service economy.

The selected companies belong to the sub-sectors of commerce, administrative activities and public administration and defence, which allows for diverse representation within the service sector. The selection of these companies was carried out through a purposive sampling process based on the accessibility and willingness of the companies to participate in the study. Regarding the distribution of questionnaires, invitations to complete an online questionnaire were sent to all employees via the institutional mail of companies that agreed to participate in the study.

A total of 615 questionnaires were distributed, with a response rate of 337 (54.79%). This relatively low response rate can be attributed to factors such as the possible workload of employees and the lack of incentives to complete the survey. Nevertheless, four questionnaires (0.65%) were excluded from the data analysis for various reasons, such as incomplete data, multiple responses to the same item and no response. Consequently, the final sample for analysis comprised 333 employees aged 18–64 years with a mean age of 38 years (SD = 2.8). In terms of tenure, employees had been with the companies for periods ranging from less than 1 year to 25 years. Specifically, 56.5% of the employees had been with the company for less than 1 year, 13.5% had been with the company between 1 and 3 years, 10.8% had been with the company between 4 and 6 years, and the remaining employees had been with the company between 7 and 25 years. In terms of gender distribution, there was a slight predominance of women in the sample (55.6% female; 44.4% male).

### 3.2. Measures

For the measuring instruments, the following standardised questionnaires were used for the measurement instruments.

#### 3.2.1. Authentic Leadership

To assess authentic leadership, this study used the Spanish version of the Authentic Leadership Questionnaire [31,62]. This instrument consists of 16 items and is structured around four key dimensions: self-awareness (the leader’s awareness of how his or her behaviour may affect others); relational transparency (the extent to which the leader presents himself or herself authentically to others); balanced processing (the leader’s ability to objectively set goals and rationally analyse data before making decisions); and internalised morality (the leader’s self-regulation of behaviour in accordance with personal values and principles, especially in the face of external pressures). After analysis, we decided to remove one item from this scale (AL1 ‘My manager says exactly what he means’) because its factor loading was below 0.6, thus validating the remaining items in the scale. Responses were collected using a Likert scale ranging from 1 (never) to 5 (always or almost always). The internal consistency (Cronbach’s alpha) of the scale was 0.951, while the composite reliability was 0.957.

#### 3.2.2. Work Autonomy and Meaningful Work (MW)

To measure work autonomy (WA) and meaningful work (MW), we used the dimension of Spreitzer’s Psychological Empowerment Instrument [63], which we adapted into Spanish [64]. The WA dimension comprises four items and assesses the extent to which an individual perceives that he or she has control over his or her work environment and can act independently in making work-related decisions. The MW dimension consists of three items and assesses the relevance and value that an employee attaches to his/her job, with reference to his/her personal norms and beliefs. Participants responded on a 7-point scale, ranging from 1 (very little) to 7 (too much). The internal consistency of the scale was 0.895 and 0.798, respectively. On the other hand, the composite reliability values of both dimensions were 0.928 and 0.882, respectively.

#### 3.2.3. Practised Creativity

To assess the practised creativity variable, we used the practised creativity dimension of the Creative and Potential Creativity Scales [41], which were adapted for the Spanish population [65]. This scale consists of five items and assesses the extent to which individuals actively engage in creative behaviours in their work environment. Response options were provided on a Likert scale ranging from 1 (strongly disagree) to 5 (strongly agree). The practised creativity scale obtained a Cronbach’s alpha reliability of 0.848 and a composite reliability value of 0.892.

#### 3.2.4. Intrapreneurial Behaviour

Finally, the intrapreneurial behaviour variable was assessed using the intrapreneurial behaviour scale [66] and adapted into Spanish [67]. This 7-item scale is divided into two subscales: innovation and risk-taking. However, after analysing the data, we decided to exclude item IB4 (i.e., I find new ways of doing things), whose factor loading did not reach the threshold of 0.6, to check the validity of the remaining items of the scale. All items were measured on a scale from 0 (strongly disagree) to 4 (strongly agree). The internal consistency of the scale was 0.856. In addition, the composite reliability of the dimensions was 0.892.

## 4. Data Analysis

In order to test each hypothesis, we performed various statistical analyses.

First, we used SPSS© v.25 Statistical software for a descriptive analysis, including measures of central tendency, dispersion and asymmetry. Additionally, as described in the previous section, we verified the reliability of the instruments through Cronbach’s alpha coefficients and composite reliability, as well as the convergent validity through the average variance extracted (AVE) and the discriminant validity (see Table 1) through the proposed criterion [68].

Partial least squares structural equation modelling (PLS-SEM) was then used to determine the variance explained and the relationships between the endogenous variables in the model. We selected this method because PLS-SEM is a well-established and efficient technique for both theory building and predictive applications [69]. It allows testing of multi-mediation models by testing one or more mediators at a time, reflecting both the statistical testing of individual mediation effects and the measurement error of the research model [70].

This analysis allowed us to estimate a predictive–explanatory study, the effect size, and the statistical significance of the coefficients associated with each pathway that constitute the proposed model [71].

## 5. Results

In order to test our working hypotheses, we carried out various analyses of the obtained data. 

First, we examined the factorial structure and loadings of each item from the questionnaires used in the study. After the analysis, it was decided to eliminate one item from the intrapreneurship behaviour variable (IB4 ‘I find new ways of doing things’) and one from authentic leadership (AL1 ‘My manager says exactly what he means’) because their factorial loadings were lower than 0.6 [68], thus validating the remaining items with their respective constructs.

We then analysed the reliability of the final items. As shown in Table 1, all constructs showed robust reliability levels, as indicated by Cronbach’s alpha and composite reliability (CR). In each case, the values exceeded the threshold of 0.7 [72]. We also examined the presence of common method bias, a phenomenon common in studies using PLS-SEM due to the measurement technique, which can affect the relationship between constructs as well as the validity of the study conclusions [70]. Collinearity, which is indicative of CMB, is considered significant when VIF values exceed 5. However, the VIF coefficients in our study remain below this threshold, confirming that our proposed model does not suffer from collinearity problems.

In addition, the average variance extracted (AVE) was analysed to assess the convergent validity of the proposed model [73]. Convergent validity assesses whether different items of a construct measure the same underlying dimension and, as such, should be highly correlated. The data presented in Table 1 confirm that all five instruments achieved predictive values greater than 50% of variance.

To assess the discriminant validity of the instruments under consideration, we applied the Fornell and Larcker’s criteria, where the square root of the AVE for each construct must exceed the correlations existing between that construct and all other constructs in the model. In Table 2, the items on the main diagonal (highlighted in bold) represent the square root of the AVE for each construct, while the off-diagonal elements represent the correlations between constructs.

After verifying the reliability and validity of the instruments and the intercorrelation between each variable, a structural model analysis was performed. We used the path coefficient (β) to determine the contribution of each predictor variable to the endogenous variable. In addition, the R-squared values were used to assess the explanatory power of the model. Bootstrap resampling was performed on 10,000 cases to test the proposed hypothesis. The results of the hypothesis testing are presented in Table 3 and in Figure 2.

From this table, all the working hypotheses were significantly confirmed, except for two hypotheses. Firstly, regarding Hypothesis 1 (AL→IB), although we obtained a significant relationship, the effect was negative and contrary to our initial hypothesis, so it was not confirmed. Specifically, high levels of authentic leadership led to a decrease in intrapreneurial behaviour, contrary to our initial postulation of a positive relationship between the two variables. Furthermore, Hypothesis 3 (AL→WA→IB) could not be confirmed either, so that job autonomy does not have a significant mediating effect.

However, the data obtained from the analyses indicate two significant relationships between the variables considered. Specifically, contrary to the previous hypothesis, PC and MW exert a significant mediating effect on the relationship between AL and IB, so although AL may directly reduce employees’ tendency to be intrapreneurs, this negative effect can be counteracted if employees find ways to practise creativity and if their work has a deep meaning for them, thus confirming Hypotheses 2 and 4.

We also assessed the effect size of the variables in relation to the hypotheses (f2), using the threshold proposed by Cohen [74], in which values of 0.02, 0.12 and 0.35 indicate small, medium and large effect sizes (see Table 2).

Finally, Figure 2 presents a graphical representation of the relationships between each of the predictors and endogenous variables (β) within the proposed study model, as well as their respective explanatory power (R2) for each relationship.

As shown in Figure 2, all R2 values were greater than 0.27 [74], supporting the explanatory power of the model. Furthermore, we calculated the goodness of fit of the model using the standardised root mean square residual (SRMR), which was 0.065, which is below the threshold of 0.08 [75], confirming the satisfactory fit of the proposed model. In addition, the Chi-square statistic was 1717.59, indicating an acceptable fit for the sample size and complexity of the model. The discrepancy between the model-implied and observed covariance matrices (d_ULS) was 2.319, and the geodesic discrepancy (d_G) was 0.967. In addition, the normalised fit index (NFI) was 0.786, suggesting a reasonable fit given the exploratory nature of the study. 

Finally, we analysed the predictive power of the model using the PLS Predict technique with 10-fold and single replication, following the guidelines of Shmueli et al. [76]. This method is recommended because, unlike the correlation coefficient (R2), it allows us to assess both the explanatory power and the predictive capacity of the model in a dataset different from the original sample selected [76]. For a more accurate assessment, the coefficient of determination (Q2) was considered. In this study, all indicators exceeded the threshold of 0, confirming the predictive validity of the proposed model [77].

In addition, we conducted an additional data analysis regarding the tenure of employees in relation to the other variables. However, our findings (see Table 4) indicate that employee tenure does not significantly influence the variables studied (*p* > 0.05). 

## 6. Discussion 

Several conclusions can be drawn from the results of our data analysis.

In our study, we examined the variables that precede intrapreneurial behaviour, taking as a reference the theories of authentic leadership [30] and social information processing (SIP) [33] in a sample of employees in the service sector in Spain. This exercise led us to develop four working hypotheses. The empirical results of this study confirmed two of the hypotheses proposed in the research model.

One finding was the relationship between authentic leadership styles and the development of intrapreneurial behaviours. Although our initial hypothesis, in line with the aforementioned theories and other similar studies (such as that developed by Edú et al. [20]), suggested a positive relationship between the two variables, the data obtained in our study revealed a significant but negative relationship between the two variables. This suggests that authentic leadership may, under certain circumstances, inhibit the development of intrapreneurial behaviour.

Through authentic leadership theory, it can be argued that the coherence and consistency of values underpinning this leadership style may limit creative diversity within a team. On the other hand, social information processing (SIP) theory suggests that employees interpret and respond to their work environment based on social cues and information they receive from their leaders. In this context, authentic leaders, by providing clear and consistent social information, can lead to group conformity and the suppression of risk-taking, which is a crucial element in the development of intrapreneurial behaviour.

These findings are partly consistent with [78,79] in that followers may have different beliefs and values than their leaders. Therefore, encouraging followers to be authentic about their values and beliefs suggests a possible increase in value divergence rather than convergence. In any case, these findings corroborate some of the results found in other studies on different leadership styles [80,81], in which leader characteristics, including status quo seeking, also limit support for risk-taking and innovation and thus hinder the development of intrapreneurial behaviour.

However, in contrast to the previous effect, our results indicate that both practised creativity and meaningful work play a positive and significant mediating role in the relationship between authentic leadership and intrapreneurial behaviour. Consistent with SIP and authentic leadership theory, these leaders, by creating a context that fosters trust and psychological safety characterised by integrity and transparency, promote engaged creativity and meaningful work by fostering a work environment in which employees feel safe and motivated to explore new ideas and take calculated risks (inherent aspects of intrapreneurial behaviour) [82,83].

Thus, although authentic leadership may have an inhibitory effect on intrapreneurial behaviour, this relationship changes when mediated by the effect of creativity and meaningful work, highlighting the importance of developing effective indirect strategies for extra-role behaviours [84,85], such as intrapreneurship.

Finally, although the included theories and previous studies suggest that authentic leadership, by promoting autonomy, can foster extra-role behaviours [86], our results show that autonomy is not a significant mediating variable in the relationship between authentic leadership and intra-entrepreneurial behaviour. 

One possible explanation is that although authentic leadership promotes autonomy, the relationship between autonomy and intrapreneurial behaviour may be conditioned by contextual or individual factors that were not included in our study. For example, it is possible that autonomy is not sufficient on its own to promote intrapreneurial behaviour without clear leadership and support for innovation. This suggests the need to develop a more nuanced understanding of how autonomy interacts with other aspects of leadership and organisational context that influence intrapreneurial behaviour.

## 7. Implications and Contributions

This study contributes to the expansion of the antecedents of intrapreneurial behaviour with particular reference to the importance of the leader in the development of this behaviour, which has several theoretical and practical implications.

Regarding the theoretical implications, the results of this study show that the relationship between both constructs (leadership and intrapreneurship) is complex and contextual, suggesting a reconsideration of authentic leadership theory, such that the implementation of this leadership style should be calibrated to foster innovation without inducing group conformity. In addition, our study contributes to the existing literature by identifying and examining the mediating role of other variables in this relationship, such as creativity and meaning at work. In doing so, we extend authentic leadership theory by testing its impact on intrapreneurial behaviour through indirect mechanisms and identify both as antecedents of intrapreneurship.

The results of this research also have practical implications for organisations and their leaders. In order to apply authentic leadership theory in a way that does not negatively impact intrapreneurial behaviour, it would be useful to consider strategies that enable authentic leaders to foster a balance between promoting consistent values and nurturing an inclusive environment that values and supports innovation and diversity of thought. This could involve adopting leadership practises that encourage the expression of divergent ideas and experimentation so that authenticity does not translate into rigidity but coexists with openness to new perspectives and flexibility.

## 8. Limitations and Future Recommendations

The study identified some limitations that should be considered by researchers.

A major limitation is the sample size and specificity, which may have affected the generalisability of our findings. Although we have attempted to address this limitation through the use of complex statistical tools, we suggest that the scope of the research be broadened to include regions and economic sectors in future research. 

A second limitation is related to the survey design, in which data were collected at a point in time that could introduce causality or endogeneity issues. When conducting cross-sectional or even traditional longitudinal studies, it is important to bear in mind that research constructs may not remain stable over time. Therefore, to draw conclusions and account for these fluctuations, we suggest further longitudinal studies that establish causality and consider both positive and negative aspects. In addition, future research should distinguish between intra- and interpersonal levels using multi-level analysis to provide a more nuanced understanding of the phenomena under investigation.

On the other hand, the results obtained may be influenced by other variables that have been shown to have a moderating effect between the type of leadership exhibited by the leader and the intra-entrepreneurial behaviours of the employees. These variables include the organisational support perceived by the workers [87] and the leader–member exchange perception (LMX) [88].

Finally, a large body of literature exists on leadership models, leading to an ongoing debate on the appropriateness of each model. Therefore, we suggest that future research should incorporate other leadership styles, seeking an integrated approach that allows for a more comprehensive understanding of the interaction between the relational and cognitive variables examined in this study of intrapreneurial behaviour.

## Figures and Tables

**Figure 1 behavsci-14-00705-f001:**
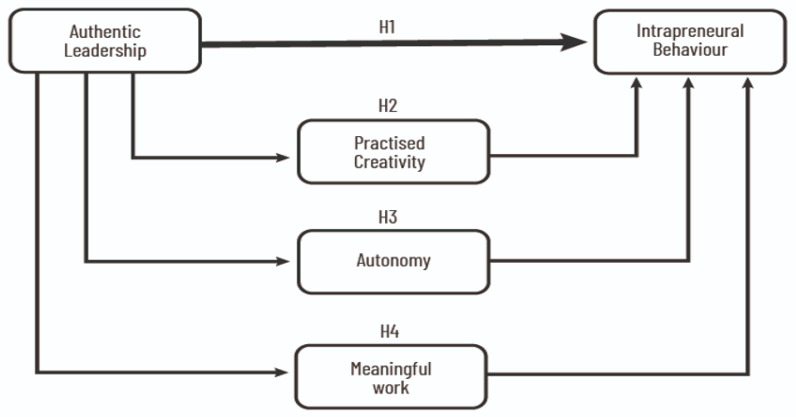
Research model..

**Figure 2 behavsci-14-00705-f002:**
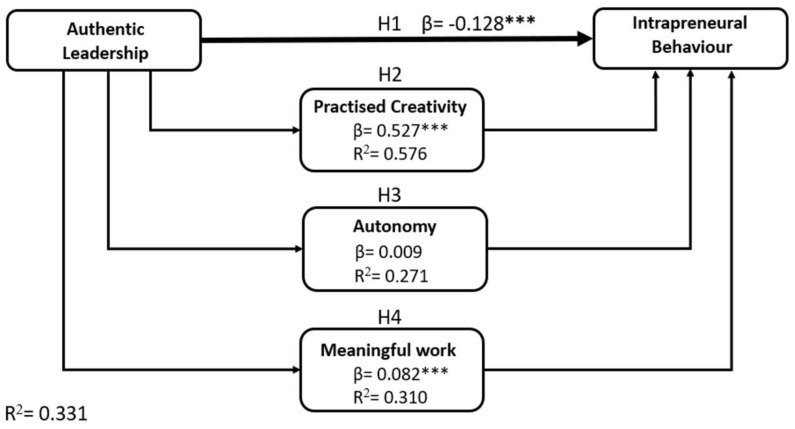
Structural model. *** *p* = 0.001.

**Table 1 behavsci-14-00705-t001:** Outler loadings, AVE, Cronbach’s alpha, and CR.

Variables	Items	Outer Loading	VIF	AVE	Cronbach’s Alpha	CR	
Authentic leadership	AL_2	0.774	2.432	0.598	0.951	0.957	
	AL_3	0.841	3.524				
	AL_4	0.704	1.902				
	AL_5	0.691	1.894				
	AL_6	0.816	3.003				
	AL_7	0.635	1.817				
	AL_8	0.778	2.316				
	AL_9	0.826	2.894				
	AL_10	0.779	2.499				
	AL_11	0.723	2.193				
	AL_12	0.853	4.599				
	AL_13	0.880	2.023				
	AL_14	0.653	3.143				
	AL_15	0.829					
	AL_16	0.770	2.446				
Work autonomy	WA_1	0.825	2.434	0.763	0.895	0.928	
	WA_2	0.837	3.072				
	WA_3	0.894	4.072				
	WA_4	0.933	1.996				
Practised creativity	PC_1	0.842	2.291	0.624	0.848	0.892	
	PC_2	0.826	2.123				
	PC_3	0.685	1.498				
	PC_4	0.765	1.630				
	PC_5	0.823	1.993				
Meaningful work	MW_1	0.876	1.975	0.714	0.798	0.882	
	MW_2	0.889	2.227				
	MW_3	0.764	1.456				
Intrapreneurial behaviour	IB_1	0.739	1.660	0.580	0.856	0.892	
	IB_2	0.796	1.917				
	IB_3	0.714	1.923				
	IB_5	0.726	2.070				
	IB_6	0.755	2.168				
	IB_7	0.832					

Note: CR = composite reliability; AVE = average variance extracted.

**Table 2 behavsci-14-00705-t002:** Means, standard deviation and correlation.

	Fornell-Larcker						
	M	SD	WA	PC	IB	AL	MW
Work Autonomy (WA)	5.19	1.165	**0.873**				
Practised Creativity (PC)	3.72	0.787	0.683 **	**0.790**			
Intrapreneurial Behaviour (IB)	3.74	0.677	0.385 **	0.489 **	**0.761**		
Authentic Leadership (AL)	3.46	0.949	0.489 **	0.486 **	0.249 **	**0.774**	
Meaningful Work (MW)	5.51	0.914	0.338 **	0.537 **	0.498 **	0.338 **	**0.845**

Note: Square root of AVE on diagonal; correlations between constructs are shown below the diagonal; ** *p* < 0.01.

**Table 3 behavsci-14-00705-t003:** Validation of research hypothesis.

Hypothesis	Coefficient	CI	*p* Values	T Statistics	F^2^	Sig
**Direct effects**						
**H1: AL→IB**	−0.128 **	(−0.249; −0.001)	0.041	2.040	0.015	Yes
**Indirect effects**						
**H2: AL→PC→IB**	0.139 **	(0.089; 0.193)	0.000	5.230		Yes
**Al→PC**	0.343	(0.267; 0.419)	0.000	8.77		Yes
**PC→IB**	0.407	(0.233; 0.544)	0.000	5.76		Yes
**H3: AL→WA→IB**	−0.131	(−0.072; 0.087)	−0.227	1.085		No
**AL→WA**	0.525	(0.451; 0.596)	0.000	13.96		Yes
**WA→IB**	0.018	(−0.139; 0.164)	0.822	0.224		No
**H4: AL→MW→IB**	0.059 **	(0.046; 0.126)	0.166	2.737		Yes
**AL→MW**	0.253	(0.151; 0.358)	0.000	4.789		Yes
**MW→IB**	0.324	(0.207−0.433)	0.000	5.652		Yes

Note: AL = authentic leadership; IB = intrapreneurial behaviour; PC = practised creativity; MW = meaningful work; WA = work autonomy. ** *p* < 0.01.

**Table 4 behavsci-14-00705-t004:** Employee tenure (ET) influence.

	Original Sample (O)	Sample Mean (M)	Standard Deviation (STDEV)	T Statistics (O/STDEV)	*p* Values
Employee tenure (ET)→IB	−0.014	−0.013	0.046	0.308	0.758
AL→Employee tenure (ET)	−0.047	−0.047	0.054	0.875	0.382
Employee tenure (ET) × AL→IB	0.034	0.037	0.056	0.612	0.541
Employee tenure (ET) × AL→PC	−0.044	−0.044	0.040	1.111	0.267
Employee tenure (ET) × AL→S	−0.034	−0.036	0.052	0.658	0.511
Employee tenure (ET) × AL→WA	−0.067	−0.067	0.041	1.622	0.105
Employee tenure (ET) × MS→IB	−0.027	−0.022	0.057	0.473	0.636
Employee tenure (ET) × WA→IB	−0.008	−0.007	0.077	0.108	0.914
Employee tenure (ET) × PC→IB	−0.005	−0.009	0.083	0.058	0.954

## Data Availability

Data are contained within the article.

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
