# Peer review of "Effects of Authentic Leadership on Intrapreneurial Behaviour: A Study in the Service Sector of Southern Spain"

_behavsci, 2024, doi:10.3390/bs14080705_

Round 1

Reviewer 1 Report (New Reviewer)

Comments and Suggestions for Authors

The paper provides interesting evidence about the impacts of authentic leadership on intrapreneurial behavior, i through the mediating roles of creativity and meaningful work. However, there are still some matters that require further clarification and improvement. In an effort to improve the paper's overall quality and comprehensiveness, I urge the authors to take the comments provided below into account and make the necessary revisions.

(1) At the beginning of the introduction, please briefly explain what is “entrepreneurship” so that the authors who are not familiar with this concept to understand it meanings.

(2) On Page 2, “Given its implications for development and  growth within organisations and its inherently autonomous and individualistic nature, both scholarly and managerial literature have exhibited burgeoning attention towards the  investigation of mechanisms to foster it, as well as the variables that can engender such an orientation among employees”, please mention what are some examples of “variables” you refer to.

(3) Similarly, for the sentence “Therefore, our aim in this study is to analyse these variables and propose an integrated model of IB”, please mention what are the “variables”.

(4) The content at the end of the introduction should be reorganized. Before mentioning the objectives of the paper, the authors must clearly identify the gap in literature. After the objectives are presented, the authors must clarify how the findings of their research make contributions.

(5) The 2 paragraphs on page 2 line 75-85 could be joined into one paragraph.

(6) Literature review on “Authentic leadership” still need to be expand by including the papers that show the recent development of this concept’s benefits. In particular, there are some more recent papers that need to be incorporated as the additional references to support the impact of Authentic leadership. I urge the authors to consider the follow papers as the additional supports.

The Impact of Authentic Leadership on Reducing Perceived Workplace Exclusion: The Moderating Roles of Collectivism and Power Distance Orientation in a Workplace, Journal of Logistics, Informatics and Service Science, 10(3), 28-46. DOI:10.33168/JLISS.2023.0303

Corporate social responsibility during unprecedented crises: the role of authentic leadership and business model flexibility", Management Decision, Vol. 58 No. 10, pp. 2213-2233. DOI:10.1108/MD-08-2020-1073

Authentic leadership and employees’ emotional labour in the hospitality industry", International Journal of Contemporary Hospitality Management, Vol. 32 No. 2, pp. 797-814. DOI: 10.1108/IJCHM-12-2018-0952

(7) The 2 paragraphs on page 3 line 125-130 could be joined into one paragraph.

(8) The hypotheses are developed solely on authentic leadership. However, there could be some other theory to be integrated to enhance the process by which authentic leaders influence behaviors of employees. Social information processing theory can be the good option.

Author Response

In the attached document you can see all the changes that have been made.

Reviewer 2 Report (New Reviewer)

Comments and Suggestions for Authors

The article provides interesting and valuable information on the role of authentic leadership in intrapersonal entrepreneurial behaviour. Below are comments to be taken into account when resubmitting a revised manuscript. These corrections will improve the clarity, reliability and scientific value of the article, making it more useful for future research in the field of intrapersonal behaviour and authentic leadership.

Purpose and Contribution to Science

The authors satisfactorily indicate the purpose of the article and its contribution to science by highlighting the importance of intrapersonal entrepreneurial behaviour for the survival of companies. However, the subject and research problem are insufficiently outlined, making it difficult to fully understand the context of the study.

Methodology and Hypotheses

Section two of the article correctly explains the variables studied and formulates the research hypotheses. The authors adopt a quantitative approach, using a questionnaire to which 333 employees responded. The results show that authentic leadership has a negative direct effect on intrapersonal behaviour, but positively supports it through practised creativity and meaningful work. Mediation of autonomy was not significant.

Research Procedure

The Sample and Data Collection section needs considerable elaboration. There is a lack of explanation of the procedure of the research conducted, including the selection of companies, the service sector and the region of Spain. It is also not clear how the questionnaires were distributed and why a low response rate was obtained. These gaps significantly reduce the credibility of the survey.

Survey Tools

The survey tools are described correctly, but an appendix with all items together with Loading values would be of added value. The Descriptive statistics and Pearson correlation sections between the study variables are also missing, which limits a full understanding of the results.

Validation and Model Analysis

The authors describe discriminant validity, but a description of Common method variance, confirmatory factor analysis and reliability assessment is missing. In addition, the 'Structural model assessment' section should include a detailed description of all indicators used, preferably presented in tables.

Discussion and conclusions

The "Discussion" section should be expanded to include a comparison of the results with previous studies by other authors. The current form is too general and does not take into account the context of the literature, making it difficult to assess the innovation and relevance of the study.

Conclusions and Recommendations

The article requires significant methodological and analytical refinement to fully meet academic standards. The authors should focus on a detailed description of the research procedure, the provision of complete statistical data and a deeper comparison with the literature.

Suggestions for Improvement

1 Reformulation of the title: The title should reflect the leadership style under study, e.g. "Towards a model for the development of intrapersonal entrepreneurial behaviour: The role of authentic leadership'.

2 Detailed description of the research procedure: Explanation of the selection of companies, sector and region, and the procedure for distributing the questionnaires.

3 Full statistics: To include sections on 'Descriptive statistics', 'Pearson correlation', 'Common method variance', confirmatory factor analysis and reliability assessment.

4 Expansion of the 'Discussion' section: Addition of comparisons with previous studies and discussion of the innovation of the study.

Author Response

In the attached document you can see all the changes that have been made.

Reviewer 3 Report (New Reviewer)

Comments and Suggestions for Authors

The paper is interesting and well written.

The analysis of the literature is meticulous without being excessively long, and it is inviting to read. I also appreciated that it was not divided into subsections to avoid interrupting the flow of thoughts.

The analyses are appropriate and correctly described.

The discussion offers interesting insights, especially since two hypotheses were not confirmed and the authors correctly proposed alternative explanations.

The limitations of the research have all been acknowledged.

I have only a few suggestions for modification to further improve the authors' work.

First, precisely because of the cross-sectional design of the research that prevents testing causal relationships (as written by the authors themselves among the limitations), it is necessary to opt for a verb different from "influence." I refer to line 56, Hypothesis 1 in line 123, and wherever it is repeated elsewhere in the text.

In the analysis of the literature, in my opinion a definition of authentic leadership should precede an explanation of the links with the other variables in the literature. I therefore recommend anticipating what is written in lines 103-104.

In the sample description, would it be possible to report the percentages of the sample by tenure ranges? 

Finally, I recommend including additional fit indices for the model tested (chi-square, CFI, TLI), in line with custom regarding SEM models.

Comments on the Quality of English Language

Some revisions are needed for grammar.

Author Response

In the attached document you can see all the changes that have been made.

Round 2

Reviewer 1 Report (New Reviewer)

Comments and Suggestions for Authors

The authors did a satisfactory job revising the paper. All the comments have been clearly handled. The overall quality of the paper is adequate for the publication. There is no additional comment from my side.

Author Response

We thank the reviewers for their valuable suggestions. 

Reviewer 2 Report (New Reviewer)

Comments and Suggestions for Authors

After the authors' revisions, I would like to express my satisfaction with the changes made. The changes made have strengthened both the clarity and scientific value of the article, and the authors have demonstrated professionalism and commitment to improving their work. The article in its current form presents a significant advance over its earlier version, and I believe it is ready for publication.

Author Response

We thank the reviewers for their valuable suggestions. 

This manuscript is a resubmission of an earlier submission. The following is a list of the peer review reports and author responses from that submission.

Round 1

Reviewer 1 Report

Comments and Suggestions for Authors

Thank you for the opportunity to review this interesting paper. In my opinion the paper would benefit from a revision. Some comments below:

- The paper starts talking about IB as its focus. However, then hypotheses concentrate on other variables including leadership, being IB one dimension where leadership and these other variables have an impact. This is confusing. The main issue in my view is that you are separating the model to create hypothesis for each relationship. However, you are referring to mediations/moderations where the key outcome is IB. What is it you are interested to test? Hypothesis should refer to the model of relationships (z mediates the relationship between x and y). Any other presentation of hypothesis is confusing and out of focus.

- Is there any reason you used PLS-SEM instead of traditional SEM? limitations of PLS-SEM? Why is relevant in this study? 

- In the result section you show the full sect of factorial analysis. Is this any relevant to your paper? any analysis to make sure that your measures work should be included in the methodology where the measure is being presented. The way to do so is by indicating the key reliability, validity, loadings range, and fit indexes in a bracket (tables with loadings are not needed). The result section should focus on your model and hypothesis. 

Formal elements:

- I would suggest standardising the way key concepts are presented. For example, you indicate intrapreneurship behaviour (IB) twice including the (IB). The second time you use capitals in the full name.

Reviewer 2 Report

Comments and Suggestions for Authors

Please, see the attached document

Comments on the Quality of English Language

I have not comments about the quality of English.

Reviewer 3 Report

Comments and Suggestions for Authors

Dear authors, thank you very much for your submission to Behavioral Sciences and for giving me the opportunity to review your manuscript reporting on the effects of authentic leadership on intrapreneurial intention. In the following, I report on several issues that came to my mind when reading your manuscript:

  1. Please make sure you use either intrapreneurial intention or intrapreneurial behavior as your dependent variable. Both is not the same.
  2. It is somewhat uncommon to present the hypotheses in the introduction. I recommend to split this long section into “Introduction” and “Theoretical Background and Hypothesis Development”.
  3. I think the choice of your first independent variable, authentic leadership, needs better justification. When I think about intrapreneurial behavior, authentic leadership is not something that come up in my mind immediately.

I hope you find my comments helpful. Good luck with your revision!

Comments on the Quality of English Language

Minor editing of English language required
